# Efficacy of Time-Restricted Eating and Behavioral Economic Intervention in Reducing Fasting Plasma Glucose, HbA1c, and Cardiometabolic Risk Factors in Patients with Impaired Fasting Glucose: A Randomized Controlled Trial

**DOI:** 10.3390/nu15194233

**Published:** 2023-09-30

**Authors:** Unyaporn Suthutvoravut, Thunyarat Anothaisintawee, Suparee Boonmanunt, Sarunporn Pramyothin, Sukanya Siriyothin, John Attia, Gareth J. McKay, Sirimon Reutrakul, Ammarin Thakkinstian

**Affiliations:** 1Department of Family Medicine, Faculty of Medicine, Ramathibodi Hospital, Mahidol University, Bangkok 10400, Thailand; unyaporn.sut@mahidol.ac.th; 2Department of Clinical Epidemiology and Biostatistics, Faculty of Medicine, Ramathibodi Hospital, Mahidol University, Bangkok 10400, Thailand; suparee.boo@mahidol.edu (S.B.); sukanya.sii@mahidol.edu (S.S.); ammarin.tha@mahidol.edu (A.T.); 3Medical Services Division, Faculty of Medicine, Ramathibodi Hospital, Mahidol University, Bangkok 10400, Thailand; sarunporn.pra@mahidol.ac.th; 4School of Medicine and Public Health, University of Newcastle, Callaghan, NSW 2308, Australia; john.attia@newcastle.edu.au; 5Centre for Public Health, School of Medicine, Dentistry, and Biomedical Sciences, Queen’s University Belfast, Belfast BT9 7BL, UK; g.j.mckay@qub.ac.uk; 6Division of Endocrinology, Diabetes, and Metabolism, Department of Medicine, University of Illinois Chicago, Chicago, IL 60612, USA; sreutrak@uic.edu

**Keywords:** behavioral economic intervention, time-restricted eating, impaired fasting glucose, cardiometabolic risk, randomized controlled trial

## Abstract

This randomized controlled trial is aimed at assessing the efficacy of combining time-restricted eating (TRE) with behavioral economic (BE) interventions and comparing it to TRE alone and to the usual care for reducing fasting plasma glucose (FPG), hemoglobin A1c (HbA1c), and other cardiometabolic risk factors among patients with impaired fasting glucose (IFG). Seventy-two IFG patients aged 18–65 years were randomly allocated for TRE with BE interventions (26 patients), TRE alone (24 patients), or usual care (22 patients). Mean FPG, HbA1c, and other cardiometabolic risk factors among the three groups were compared using a mixed-effect linear regression analysis. Mean body weight, FPG, HbA1c, fasting insulin, and lipid profiles did not significantly differ among the three groups. When considering only patients who were able to comply with the TRE protocol, the TRE group showed significantly lower mean FPG, HbA1c, and fasting insulin levels compared to the usual care group. Our results did not show significant differences in body weight, blood sugar, fasting insulin, or lipid profiles between TRE plus BE interventions, TRE alone, and usual care groups. However, TRE might be an effective intervention in lowering blood sugar levels for IFG patients who were able to adhere to the TRE protocol.

## 1. Introduction

Diabetes mellitus (DM) is a major health problem in Thailand with a reported prevalence of 9.9% in 2014 [1]. Diabetes is a significant risk factor for cardiovascular diseases (CVDs), chronic kidney disease (CKD), peripheral arterial disease, and diabetic retinopathy which together account for 4% of mortality in the Thai population [2]. Impaired fasting glucose (IFG) is an intermediate stage between normal blood sugar and DM. Individuals with IFG have a 13 times higher risk of DM than individuals with normal blood glucose [3]. Therefore, intensive lifestyle modification by diet control and exercise is an important intervention to reduce the progression towards DM in this population [4]. Caloric restriction (CR) through low and very low-calorie diets is an effective dietary intervention that significantly reduces the risk of DM but the adherence and sustainability of this dietary intervention over the long term are questionable [5]. As a result, other methods of diet control, such as time-restricted eating (TRE), have emerged as potential alternatives.

TRE is one type of dietary approach that limits the daily eating window, commonly to less than 10 h per day, prolonging the fasting time [6,7,8]. Evidence from animal studies found that increased fasting time could reduce free radical production, inhibit inflammation, and increase stress resistance, leading to improved metabolic health and glucose regulation [9,10,11]. To date, there have been limited randomized controlled trials (RCTs) investigating the impact of TRE in individuals with a high risk of DM, such as IFG. One crossover RCT examined the effects of early TRE (where food intake is restricted to the early period of the day) or delayed TRE (where food intake is restricted to the later period of the day) on 15 men who were at risk of type 2 diabetes mellitus. The study revealed that both early and delayed TRE improved the glycemic response, but only the early TRE was able to lower average fasting plasma glucose (FPG) levels [12]. Another RCT explored the effects of early TRE in eight IFG men and found that it could reduce insulin levels, blood pressure, and food appetite and increase insulin sensitivity and beta-cell responsiveness but could not reduce FPG [13]. Consequently, further RCTs are required to investigate the long-term effects of TRE in both male and female patients at risk of type 2 diabetes.

Despite widespread adoption and given the lack of restrictions on food groups, specific macronutrients, or constant calorie monitoring [14,15], there remains uncertainty regarding the long-term adherence to TRE in real-life scenarios. Previously, various behavioral economic interventions have been employed to enhance adherence to dietary control, including the use of financial incentives [16,17] and text reminders [18]. Behavioral economics integrates principles and methods from psychology and economics to better understand human decision-making [19,20] and motivation. Previous research has demonstrated the effectiveness of financial incentives in promoting a healthy lifestyle, such as smoking cessation, physical activity [21], and weight loss [22]. Text reminders that reinforce an individual’s commitment, performance, or goals can serve as immediate prompts to prioritize and maintain adherence to treatment [23]. Consequently, employing behavioral economic (BE) techniques, namely financial incentives and text reminders, could potentially enhance compliance with lifestyle modifications like TRE and even facilitate the long-term maintenance of behavioral changes.

Nevertheless, no studies have assessed the efficacy of combined TRE with BE interventions in patients with IFG. Therefore, this RCT was conducted to compare the efficacy of additional BE combined with TRE against TRE alone and the usual care in patients with IFG. Levels of FPG, HbA1c, and other cardiometabolic risk factors, including body mass index (BMI), systolic blood pressure (SBP), diastolic blood pressure (DBP), fasting insulin, serum triglyceride levels (TG), total cholesterol, low-density lipoprotein cholesterol (LDL-C), high-density lipoprotein cholesterol (HDL-C), and high sensitivity C-reactive protein (hs-CRP), were compared among the TRE plus BE intervention, TRE alone, and usual care groups.

## 2. Materials and Methods

### 2.1. Study Design

This parallel RCT was conducted at the outpatient clinic of the Department of Family Medicine, Faculty of Medicine, Ramathibodi Hospital, from October 2021 to February 2023. This research complies with the Consolidated Standards of Reporting Trials (CONSORT) statement. The trial protocol was registered at the Thai Clinical Trials Registry (TCTR20210520002) and was published elsewhere [24]. The protocol was approved by the Ethics Committee of Ramathibodi Hospital, Mahidol University (MURA 2021/389).

### 2.2. Participants

This study included patients and staff of Ramathibodi Hospital using the following eligibility criteria: adults aged 18 to 65 years, diagnosed with IFG (i.e., FPG of 100–125 mg/dL and HbA1c less than 6.5%) [25], having BMI ≥ 25 kg/m^2^, and willing to provide informed consent. Patients were excluded if they met any of the following criteria: (1) followed a ketogenic or vegetarian diet, (2) worked night shift for a minimum of 3 h between 10:00 PM and 5:00 AM on more than one day per week, (3) experienced body weight changes exceeding 5 kg in the three months prior to study enrolment, (4) were in receipt of medication to be consumed with food either before 8:00 AM or after 5:00 PM, (5) were pregnant or breastfeeding, (6) had psychiatric disorders, such as eating or mood disorders (except depression), (7) were taking corticosteroid or anti-diabetic medications, (8) had a history of bariatric surgery, or (9) had impaired nutrient absorption.

### 2.3. Randomization

Patients were randomly assigned to (1) TRE plus BE, (2) TRE alone, or (3) usual care in a ratio of 1:1:1. A random sequence was generated using stratified-block randomization with varying block sizes (6 and 9) and age stratification (i.e., 18–59 years and 60–65 years).

A random sequence list was concealed in sequential opaque sealed envelopes.

### 2.4. Study Interventions

This study focused on two specific interventions: TRE and BE-informed interventions. In the TRE groups, patients were instructed to restrict their daily food intake to a 9 h window (between 8:00 AM and 5:00 PM), without any limitation on the types of food and beverages consumed. This approach involved a fasting period of 15 h per day.

The BE interventions included financial incentives and text reminders aimed at increasing adherence to TRE. Participants were eligible to receive a monthly monetary payment of 1000 THB (approximately 85 USD) if they self-reported adherence to the TRE for at least 5 days a week over a 4-week period. In addition, text reminders were sent to participants every 2 days to reinforce their commitment to achieving the TRE goal of 5 days per week, track their personal performance, and remind them of the TRE interval. Participants in all three groups were also requested to maintain a logbook to record their first and last mealtime each day.

### 2.5. Outcomes

The primary outcomes were FPG and HbA1c levels measured at 12 weeks after randomization. Secondary outcomes included body weight, SBP, DBP, fasting insulin, TG, total cholesterol, LDL-C, HDL-C, and hs-CRP. The homeostatic model assessment of insulin resistance (HOMA-IR) was calculated by multiplying the fasting insulin (mIU/L) with FPG (mg/dL) and dividing by 405 to measure insulin sensitivity in the participants. FPG was measured using hexokinase glucose-6 phosphate dehydrogenase. HbA1c levels were measured using the turbid metric inhibition immunoassay certified by the National Glycohemoglobin Standardization Program. Serum triglyceride was measured by lipase/glycerol kinase glycerol-3-phosphate oxidase. HDL-C and LDL-C levels were measured by the accelerator selective detergent method. Chemiluminescence and nephelometry methods were used for measuring fasting insulin and hs-CRP, respectively. Body weight was measured without shoes to the nearest 100 g using a Seca 284 Wireless Measuring Station—Weight & Height 2841300109. Blood pressure was measured by the Connex Spot Monitor; Welch Allyn, Inc. (Auburn, NY, USA) with an automatic blood pressure cuff after resting for at least 15 min. Body weight and blood pressure were measured by trained research assistants.

All primary and secondary outcomes were assessed at baseline and weeks 4, 8, and 12 after randomization. Possible adverse events of TRE, such as syncope, dizziness, and light headedness, were monitored and recorded throughout the study period.

### 2.6. Study Procedure and Data Collection

At the first visit, potential participants were screened for the eligibility criteria (e.g., FPG, HbA1c, age, and BMI) by investigators and research assistants.

One week after enrolment (2nd visit), demographic information (e.g., age, sex, educational level, and marital status), underlying diseases (e.g., hypertension, dyslipidemia, chronic kidney disease, fatty liver disease, and history of gestational diabetes mellitus), and health risk behaviors (i.e., smoking and alcohol consumption) were collected by an interview with trained research assistants. Blood pressure, body weight, and height were measured, and participants were randomly allocated to one of three groups: TRE plus BE, TRE alone, and usual care. Physical examination including blood pressure and body weight, and laboratory measurements (i.e., FPG, HbA1c, TG, total cholesterol, LDL-C, HDL-C, fasting insulin, and hs-CRP) were obtained during the 3rd visit (4th week), 4th visit (8th week), and 5th visit (12th week) after randomization or study end. Dietary intake at baseline and weeks 4, 8, and 12 post randomization were collected using 24 h food recall. Participants were asked to record their food diary for 7 days at each visit. INMUCAL-nutrients version 4.0 (developed by the Institute of Nutrition, Mahidol University, Thailand; https://inmu2.mahidol.ac.th/inmucal/index.php, accessed on 24 May 2023) was used for calculating nutrient intake from dietary data.

### 2.7. Statistical Analysis

A superiority trial approach utilizing a one-way analysis of variance method was applied for estimating the sample size. The baseline mean and standard deviation (SD) of FPG in the prediabetes cohort were 105 and 9 mg/dL, respectively [26]. We anticipated that receiving TRE plus BE, and TRE alone would lead to a reduction in the FPG of approximately 7% and 5%, respectively, compared to the control group. With type I error and power set to 0.05 and 0.8, respectively, a total of 90 participants, with 30 participants per group, were required to detect these differences. However, given that the recruitment was adversely affected since it fell within COVID-19 restrictions, the final sample size was limited to 72 participants.

Mean (SD) or median (interquartile range, IQR) and frequency and percentages were used to describe continuous data and categorical data, respectively. Means of primary and secondary outcomes among the three groups were compared using a mixed-effect linear regression model by regressing an outcome of intervention and time, considering patients as a random effect (i.e., repeated measure at 4, 8, and 12 weeks after randomization) and the intervention arms (TRE plus BE and TRE alone versus usual care) as a fixed effect. Marginal means and differences between any pair of the three interventions were estimated accordingly. If baseline characteristics differed between the 3 groups, these were adjusted using multivariate regression analysis.

All analyses were performed using three approaches. First, an intention-to-treat analysis (ITT) was applied by considering interventions that were initially assigned regardless of compliance. Second, a per-protocol analysis (PPA) excluded patients who did not comply with TRE (i.e., complied < 5 days per week) in the TRE plus BE and TRE alone and patients in the usual care group who took TRE 5 days or more per week from the analysis. Third, an “as-treated” analysis kept participants in either TRE plus BE or TRE alone if they complied with TRE ≥ 5 days/week; otherwise, they were considered in the usual care group. Likewise, participants in a usual care group were moved to the TRE group if they adhered to TRE ≥ 5 days/week. If a participant missed an appointment in week 4, 8 or 12, the previous outcome measure was carried forward to impute the missing outcome to minimize the lost to follow-up.

All analyses were performed using STATA 18.0. A *p*-value < 0.05 was considered statistically significant.

## 3. Results

A total of 80 patients with IFG were recruited following eligibility screening from October 2021 to February 2023. Four, three, and one patients were excluded due to FPG > 125 mg/dL, FPG < 100 mg/dL, and HbA1c > 6.5%, respectively. Thus, 72 patients with IFG were randomly allocated to the TRE plus BE (n = 26), TRE alone (n = 24), and usual care (n = 22) groups. One participant in the TRE group was lost to follow-up in weeks 4, 8, and 12; thus, this patient was excluded from the analysis due to having no data available after randomization. One participant in the usual care group was lost to follow-up in week 12, while one patient in the TRE alone group was unable to attend the week 8 follow-up but was followed up again in week 12. Therefore, 71 participants were included in the ITT analysis with sample sizes of 26, 23, and 22 for TRE plus BE, TRE, and usual care, respectively (see Figure 1). Considering adherence to interventions, 12/26 (46%) in the TRE plus BE group and only 2/23 (9%) in the TRE alone, adhered to TRE ≥ 5 days/week, while 1/22 (5%) in the usual care group adhered to TRE ≥ 5 days/week. Therefore, the PPA could not be performed due to the low number of participants in the TRE alone group. For the as-treated analysis, 15 participants (12 from TRE plus BE, 2 from TRE alone, and 1 from the usual care group) were included in the TRE group, while 56 participants (14 from TRE plus BE, 21 from TRE alone, and 21 from usual care groups) were included in the usual care group.

Baseline participant characteristics are presented in Table 1. The mean age was 54.6 years (8.1); the majority were female (69.4%) and self-reported as never smokers (86.1%) or alcohol consumers (56.9%). Most participants had dyslipidemia (90.3%) and approximately half had hypertension (55.6%), while none reported a history of depression, gestational diabetes, or coronary artery disease. Demographic characteristics were not significantly different between the three groups. Baseline dietary intakes and laboratory measures were also similar. However, the amount of carbohydrates (mean = 143.3 vs. 179.3 vs. 142.9 g/day), cholesterol intake per day (mean = 189.3 vs. 201.9 vs. 207.0 mg/d), serum triglycerides (mean = 119.5 vs. 134.5 vs. 131.5 mg/dL), and fasting insulin levels (mean = 8.8 vs. 8.6 vs. 11.1 mIU/L) were clinically different among the three groups, likely due to the small sample size. Therefore, these covariates were considered as confounding factors in the multivariate mixed linear regression analysis.

### 3.1. FPG and HbA1c Levels

#### 3.1.1. Intention-to-Treat Analysis

When compared to baseline levels, FPG decreased over time for all three groups (see Figure 2A); however, a significant decrease in FPG was only found in the TRE alone group (*p*-value = 0.001). TRE plus BE and TRE alone lowered the FPG levels by −1.74 mg/dL (95% CI: −5.60, 2.12) and −3.03 mg/dL (95% CI: −7.00, 0.93), respectively, when compared to the usual care group (see Table 2), although these did not reach statistical significance. Additionally, the effect of lowering the FPG level was not significantly different between TRE plus BE and TRE groups with a mean difference (95% CI) of 1.29 (−2.46, 5.04) mg/dL; see Table 2.

The HbA1c levels after receiving interventions did not decrease significantly when compared to baseline levels in all three groups (Figure 2B). Mean differences (95% CI) in TRE plus BE vs. usual care and TRE alone vs. usual care were −0.17% (−0.38, 0.04) and −0.15% (−0.36, 0.07), respectively (see Table 2). This suggested that TRE plus BE and TRE alone did not provide any additional benefit in reducing HbA1c levels. Additionally, the HbA1c levels after receiving the interventions were not significantly different between the TRE plus BE and TRE alone groups with a mean difference of −0.02% (−0.21, 0.17).

#### 3.1.2. As-Treated Analysis

TRE significantly decreased FPG and HbA1c levels, when compared to usual care with mean differences of −4.74 mg/dL (−8.58, −0.90) for FPG and −0.24% (−0.457, −0.03) for HbA1c level (Table 3).

### 3.2. Body Weight, Nutrition Intake, Systolic and Diastolic Blood Pressure

#### 3.2.1. Intention-to-Treat Analysis

Body weight significantly decreased over time when compared to the baseline levels for all three groups (see Figure 2C). Among the intervention groups, both TRE plus BE and TRE did not additionally lower body weight when compared to the usual care, with mean differences of 0.13 kg (−5.80, 6.07) and −2.67 kg (−8.65, 3.30); see Table 2. Mean body weight was also not significantly different between TRE plus BE and TRE alone groups, with a mean difference of 2.81 kg (−0.23, 5.85). In addition, total energy, carbohydrate, protein, fat, cholesterol, and simple sugar intakes were not significantly different among the three groups (Table 2). However, the amount of saturated fat intake was significantly lower in the TRE plus BE group, compared to the usual care group.

TRE plus BE significantly decreased mean SBP by approximately −9.67 mmHg (−17.40, −1.95) and −10.36 mmHg (−17.96, −2.76), when compared to the usual care and TRE alone groups, respectively. However, these effects were not observed in DBP with the mean difference in these corresponding groups of −3.42 mmHg (−7.70, 0.87) and −3.78 mmHg (−8.03, 0.47) vs. usual care. Mean SBP and DBP levels in the TRE alone group were not significantly different from those in the usual care group (Table 2).

#### 3.2.2. As-Treated Analysis

Similar to ITT, the TRE did not decrease body weight when compared to usual care. However, total energy, total fat, saturated fat, and simple sugar intakes were significantly lower in TRE than in the usual care groups (Table 3). DBP was significantly lower in the TRE group than in the usual care group (mean difference = −5.28 (−9.57, −0.99) mmHg) but not SBP (mean difference = −7.21 (−15.25, 0.83) mmHg); see Table 3.

### 3.3. Lipid Profiles

#### 3.3.1. Intention-to-Treat Analysis

TRE plus BE and TRE alone lowered serum TG, total cholesterol, and LDL-C levels when compared to the usual care group, but these effects did not reach statistical significance (Table 2). When comparing TRE plus BE and TRE alone, serum TG, total cholesterol, and LDL-C levels were not significantly different.

Regarding HDL-C, both TRE plus BE and TRE alone did not significantly improve HDL-C when compared to usual care. Moreover, the mean HDL-C was not significantly different between the TRE plus BE and TRE groups (Table 2).

#### 3.3.2. As-Treated Analysis

Results from the as-treated analysis were similar to the results from ITT analysis: Total cholesterol, LDL-C, and HDL-C levels were not significantly different among TRE and usual care groups (Table 3). However, serum TG in the TRE group was significantly lower than serum TG in the usual care group (mean difference = −18.54 mg/dL; 95% CI: −37.03, −0.05).

### 3.4. Fasting Insulin and HOMA-IR

#### 3.4.1. Intention-to-Treat Analysis

Mean differences in fasting insulin and HOMA-IR between TRE plus BE vs. usual care and TRE alone vs. usual care were −1.59 (−3.34, 0.16) and −1.57 (−3.37, 0.23) for fasting insulin and −0.40 (−0.91, 0.11) and −0.47 (−1.00, 0.05) for HOMA-IR, respectively. These suggested that both TRE plus BE and TRE alone did not additionally decrease fasting insulin and insulin resistance when compared to usual care. There was also no significant difference in fasting insulin and HOMA-IR levels between TRE plus BE and TRE alone groups (Table 2).

#### 3.4.2. As-Treated Analysis

Fasting insulin and HOMA-IR levels were significantly lower in the TRE group than in the usual care group, with mean differences of −1.94 (95% CI: −3.71, −0.18) and −0.61 (−1.12, −0.10), respectively.

### 3.5. Hs-CRP

#### 3.5.1. Intention-to-Treat Analysis

There was a significant reduction in hs-CRP in the TRE group, when compared to the usual care group with a mean difference of −1.68 (−3.20, −0.14), while there was no significant difference in hs-CRP levels between TRE plus BE and usual care groups with a mean difference = −0.43 (−1.91, 1.05). In addition, Hs-CRP levels did not differ significantly between TRE plus BE and TRE alone groups (Table 2).

#### 3.5.2. As-Treated Analysis

Contrary to ITT, the hs-CRP level was not significantly different between TRE and usual care groups, with a mean difference of −0.92 (95% CI: −2.45, 0.60).

### 3.6. Adverse Events

None of the patients in TRE plus BE, TRE alone, and usual care groups reported adverse events.

## 4. Discussion

Our study investigated the effects of TRE with and without BE on patients with IFG. The results showed that neither TRE plus BE nor TRE alone had a significant impact on FPG, HbA1c, body weight, DBP, lipid profiles, and fasting insulin levels when compared to usual care. However, when comparing TRE plus BE to usual care, there was a significant reduction in SBP. Additionally, when comparing TRE alone to usual care, a significant decrease in hs-CRP levels was also observed. When considering the “as-treated” analysis, complying with the TRE protocol brought significant improvements in FPG, HbA1c, serum TG, fasting insulin, HOMA-IR, and DBP levels when compared to usual care alone.

### 4.1. Effect of TRE on Blood Sugar Level and Cardiometabolic Risk Factors

The effectiveness of TRE in improving blood glucose levels, weight reduction, and lipid profiles in patients with IFG has been investigated previously. A recent RCT conducted on IFG patients found that TRE with a 16 h fasting period for three weeks led to a significant decrease in FPG levels at the three-month follow-up [27]. However, the findings from multiple studies in patients with obesity/overweight [28,29,30], non-alcoholic fatty liver disease [31], or prediabetes [13], did not show significant benefits of TRE in lowering FPG levels when compared to the designated control group. Our results are consistent with these findings, as both TRE plus BE and TRE alone did not significantly lower FPG and HbA1c levels when compared to usual care. The lack of significant results might be attributed to low adherence to the TRE regimen among our participants, as well as our low sample size.

Our hypothesis was confirmed when analyzing the data considering adherence (as-treated analysis); TRE showed significant improvements in FPG, HbA1c, fasting insulin, and HOMA-IR levels compared to usual care. Similar results were observed previously in a crossover RCT where prediabetes patients who adhered to TRE for five weeks experienced a significant reduction in fasting insulin levels [12]. Findings from both our study and that in prediabetes suggest that adhering to TRE may enhance insulin sensitivity and β-cell responsiveness in prediabetes patients, potentially reducing the risk of progression to DM for these individuals.

Body weight changes in our study were not significantly different among the three groups. However, by this study’s end, both the TRE plus BE and TRE alone groups showed a significant decrease in body weight compared to their baseline measurements. Our results support previous findings from individuals with prediabetes, which showed no significant difference in body weight reduction between the TRE and control groups. However, in both groups, there was a notable loss of body weight compared to their respective baseline measurements [12]. Similar positive outcomes related to body weight were observed in various RCTs involving patients with obesity or overweight conditions. These studies reported a significant decrease in body weight when participants followed a TRE approach compared to their baseline measurements [28,32,33]. The beneficial effect of TRE on weight loss could potentially be attributed to the unintentional reduction in energy intake during prolonged fasting periods [34]. Moreover, extending the fasting window increases the utilization of stored fat, leading to higher levels of ketone bodies, which in turn can suppress appetite and result in reduced energy intake [35].

Weight reduction can reduce future diabetic risk in individuals with prediabetes and obesity. Our study observed a significant reduction in body weight from the baseline in participants who adhered to the TRE protocol.

Consequently, it is essential to consider that the observed benefits of TRE in reducing FPG and HbA1c, as identified from the “as-treated” analysis, might be attributed to mechanisms other than weight loss. While weight reduction is well-accepted for its preventive role in type 2 diabetes onset and progression, it appears that TRE may exert its positive effects on FPG and HbA1c through different pathways, independent of weight reduction. Glucose metabolism is intricately regulated by the circadian clock. Aligning the eating period with the body’s natural circadian rhythms, such as consuming meals earlier in the day and fasting for the remaining time, can have a significant impact on metabolic processes. This eating pattern has been shown to restore cyclic AMP response element-binding protein phosphorylation, which, in turn, leads to a reduction in gluconeogenesis and contributes to improved glucose homeostasis [34].

Our study observed significantly lower SBP in the TRE plus BE group, but no significant change was found in the TRE alone group. An RCT conducted on men with prediabetes also reported significant decreases in both SBP and DBP compared to the control group [13]. However, findings from other RCTs involving diverse population groups showed inconsistent effects of TRE on blood pressure levels [33,36,37,38]. A systematic review analyzing the effects of TRE with different feeding windows suggested that the benefits of TRE might be more pronounced in early TRE, where food intake is restricted to the early period of the day [39], likely due to better alignment with circadian rhythms.

The potential benefits of TRE in suppressing inflammation, identified from animal studies [9,10,11], have yet to be fully recognized in humans. In our study, we observed a significant decrease in hs-CRP levels in the TRE alone group, but not in the TRE plus BE group. This contradicts previous findings [13,40], which reported that TRE did not significantly affect hs-CRP levels. However, the beneficial effects of TRE in improving hs-CRP levels in our study were not observed in the “as-treated” analysis. This raises the possibility that our observed results may have occurred by chance, emphasizing the need for further studies to confirm the potential impact of TRE on the inflammatory process.

### 4.2. Effect of Behavioral Economic Interventions on Adherence to TRE Protocol

Despite the proposed seemingly easier adoption of the TRE approach compared to other diet-control techniques, adherence to TRE remains challenging as observed by the low adherence rate in the TRE alone group (9%). Applying behavioral-economic informed interventions (i.e., financial incentives and text reminders in this study) could motivate some participants to higher adherence, as observed in the TRE plus BE group (46%) compared to TRE alone (9%). This may be due to the short-term financial incentives that complement future health benefits [17], while reminders make TRE salient to participants every two days [41]. However, we could not disentangle the effects of financial incentives and reminders from each other. A recent feasibility RCT comparing text reminders and text reminders plus financial incentives (in the form of a deposit contract) to increase adherence to TRE in patients with obesity and hypertension showed no difference between both groups [42], although this study was underpowered. Future studies focusing on the effect of each of these behavioral-economic informed interventions might be of interest for clinical and cost-effective intervention design.

### 4.3. Strengths and Limitations

Given the limited evidence of the efficacy of TRE in individuals with IFG, our study contributes valuable knowledge to this field. The findings from our RCT can inform the recommendation of appropriate dietary interventions to reduce future diabetic risk. Furthermore, our study assessed the efficacy of BE-informed interventions to promote intervention adherence. This evidence is valuable for encouraging adherence not only to dietary but other interventional approaches.

However, our study had several limitations. First, ongoing COVID-19 pandemic restrictions during the recruitment period resulted in a smaller sample size than anticipated which reduced the statistical power of our findings. Second, the participant adherence rate to the TRE protocol in our study was quite low. Consequently, the non-significant effectiveness of TRE in our study should be interpreted with caution given the poor intervention compliance. This is supported by the results from the “as-treated” analysis that point to potentially significant benefits of TRE for improving blood sugar and fasting insulin levels. Third, our interventions could not be blinded; hence, the findings may be subject to observer and information bias. However, the majority of our outcomes were objectively measured, and the assessors were blinded to allocation; thus, measurement error or ascertainment bias was minimized. In addition, our study did not measure oral glucose tolerant test. As a result, we were unable to determine the status of impaired glucose tolerance among the participants in the study. Lastly, only surrogate outcomes such as FPG, HbA1c, body weight, and other laboratory tests were assessed; thus, further RCTs that measure long-term or clinically important outcomes, such as DM incidence, are needed.

## 5. Conclusions

The current study demonstrated a non-significant benefit of TRE plus BE and TRE alone in improving body weight, blood sugar, fasting insulin, and lipid profiles, when compared to the usual standard of care, albeit subject to reduced statistical power. However, the benefits of TRE in reducing blood sugar, fasting insulin, and HOMA-IR levels were observed specifically among subjects who were able to adhere to the TRE protocol. This suggests that TRE may offer promise as a dietary intervention for high-risk individuals and thereby potentially decrease the associated risk of DM onset and progression. In addition, BE interventions might increase the adherence to TRE.

## Figures and Tables

**Figure 1 nutrients-15-04233-f001:**
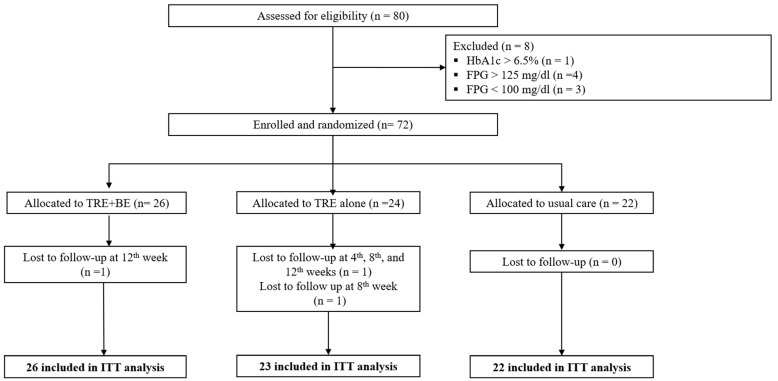
Participant flow diagram.

**Figure 2 nutrients-15-04233-f002:**
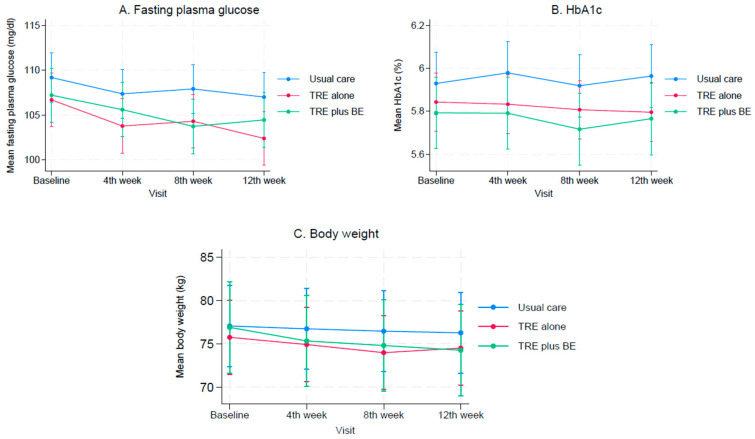
Mean changes in fasting plasma glucose, HbA1c, and body weight in time-restricted eating plus behavioral economic intervention, time-restricted alone, and usual care groups: (**A**) Fasting plasma glucose; (**B**) HbA1c; (**C**) Body weight.

**Table 1 nutrients-15-04233-t001:** Baseline characteristics of participants.

Characteristics	Total (N = 72)	TRE Plus Behavioral Economics (N = 26)	TRE Alone (N = 24)	Usual Care (N = 22)	*p*-Value
Age at enrollment, year, mean (SD)	54.6 (8.1)	53.2 (9.2)	55.5 (7.2)	55.2 (7.9)	0.558
Sex					
- Female	50 (69.4)	18 (69.2)	16 (66.7)	16 (72.7)	0.905
- Male	22 (30.6)	8 (30.8)	8 (33.3)	6 (27.3)	
Educational level					
- Primary school	14 (19.7)	6 (24.0)	4 (16.7)	4 (18.2)	0.932
- Secondary school	19 (26.8)	6 (24.0)	6 (25.0)	7 (31.8)	
- College or higher	38 (53.5)	13 (52.0)	14 (58.3)	11 (50.0)	
Marital status					
- Single	11 (15.3)	6 (23.1)	3 (12.5)	2 (9.1)	0.788
- Married	48 (66.7)	15 (57.7)	18 (75.0)	15 (68.2)	
- Divorced	10 (13.9)	4 (15.4)	2 (8.3)	4 (18.2)	
- Widowed	3 (4.2)	1 (3.8)	1 (4.2)	1 (4.5)	
Reimbursement					
- UHC	6 (8.3)	4 (15.4)	2 (8.3)	0 (0.0)	0.524
- SSS	21 (29.2)	7 (26.9)	8 (33.3)	6 (27.3)	
- Civil servant	29 (40.3)	11 (42.3)	9 (37.5)	9 (40.9)	
- Others	16 (22.2)	4 (15.4)	5 (20.8)	7 (31.8)	
Smoking status					
- Current	1 (1.4)	0 (0.0)	0 (0.0)	1 (4.5)	0.580
- Past	9 (12.5)	3 (11.5)	4 (16.7)	2 (9.1)	
- Never	62 (86.1)	23 (88.5)	20 (83.3)	19 (86.4)	
Alcohol consumption					
- Current	15 (20.8)	6 (23.1)	6 (25.0)	3 (13.6)	0.831
- Past	16 (22.2)	6 (23.1)	4 (16.7)	6 (27.3)	
- Never	41 (56.9)	14 (53.8)	14 (58.3)	13 (59.1)	
Family history of DM					
- Yes	38 (52.8)	17 (65.4)	12 (50.0)	9 (40.9)	0.226
- No	34 (47.2)	9 (34.6)	12 (50.0)	13 (59.1)	
Underlying diseases					
Hypertension					
- Yes	40 (55.6)	12 (46.2)	14 (58.3)	14 (63.6)	0.452
- No	32 (44.4)	14 (53.8)	10 (41.7)	8 (36.4)	
Dyslipidemia					
- Yes	65 (90.3)	23 (88.5)	23 (95.8)	19 (86.4)	0.515
- No	7 (9.7)	3 (11.5)	1 (4.2)	3 (13.6)	
Depression					
- Yes	0 (0.0)	0 (0.0)	0 (0.0)	0 (0.0)	NA
- No	72 (100.0)	26 (100.0)	24 (100.0)	22 (100.0)	
CKD					
- Yes	2 (2.8)	1 (3.8)	0 (0.0)	1 (4.5)	0.592
- No	70 (97.2)	25 (96.2)	24 (100.0)	21 (95.5)	
CAD					
- Yes	0 (0.0)	0 (0.0)	0 (0.0)	0 (0.0)	NA
- No	72 (100.0)	26 (100.0)	24 (100.0)	22 (100.0)	
CVA					
- Yes	2 (2.8)	0 (0.0)	2 (8.3)	0 (0.0)	0.128
- No	70 (97.2)	26 (100.0)	22 (91.7)	22 (100.0)	
NAFLD					
- Yes	8 (11.1)	2 (7.7)	2 (8.3)	4 (18.2)	0.447
- No	64 (88.9)	24 (92.3)	22 (91.7)	18 (81.8)	
History of GDM					
- Yes	0 (0.0)	0 (0.0)	0 (0.0)	0 (0.0)	NA
- No	72 (100.0)	26 (100.0)	24 (100.0)	22 (100.0)	
Cancer					
- Yes	1 (1.4)	1 (3.8)	0 (0.0)	0 (0.0)	0.408
- No	71 (98.6)	25 (96.2)	24 (100.0)	22 (100.0)	
Physical examination					
- Weight, kg, mean (SD)	76.6 (12.9)	77.3 (15.8)	75.4 (9.6)	77.1 (12.7)	0.856
- BMI, kg/m^2^, mean (SD)	29.9 (3.8)	30.3 (4.9)	29.2 (2.9)	30.3 (3.2)	0.536
- SBP, mmHg, mean (SD)	135.1 (15.3)	134.6 (14.4)	134.0 (15.3)	137.0 (16.9)	0.789
- DBP, mmHg, mean (SD)	79.9 (7.7)	80.0 (7.4)	79.2 (9.0)	80.5 (6.9)	0.835
- WC, cm, mean (SD)	95.7 (9.3)	95.8 (10.4)	94.8 (7.8)	96.4 (9.7)	0.837
- HC, cm, mean (SD)	104.4 (7.5)	104.7 (8.6)	103.8 (6.5)	104.9 (7.4)	0.864
- NC, cm, mean (SD)	37.0 (3.3)	37.3 (3.4)	37.5 (3.1)	36.3 (3.3)	0.427
Dietary intakes					
- Total energy intake, kcal/day, mean (SD)	1141.0 (522.1)	1121.5 (547.9)	1210.5 (614.1)	1095.4 (382.5)	0.746
- Carbohydrate, g/day, mean (SD)	154.5 (85.4)	143.3 (58.0)	179.3 (129.1)	142.9 (47.1)	0.259
- Protein, g/day, mean (SD)	47.4 (25.1)	47.2 (30)	46 (20.3)	49.3 (23.8)	0.911
- Total fat, g/day, median (IQR)	32.9 (19.2–46.4)	34.7 (18.3–55.1)	28.0 (21.1–43.1)	35.0 (19.0–50.0)	0.744
- Cholesterol, mg/day, median (IQR)	205.1 (105.0–335.4)	158.1 (73.2–406.7)	205.1 (134.7–283.1)	248.2 (182.1–340.8)	0.607
- Simple sugar, g/day, median (IQR)	31.9 (17.1–51.1)	32.3 (18.5–48.0)	35.0 (16.5–60.6)	29.4 (17.3–44.5)	0.338
- Saturated fat, g/day, median (IQR)	6.7 (3.3–12.2)	6.7 (3.0–14.5)	7.0 (3.1–12.2)	6.8 (3.7–11.8)	0.419
Laboratory results					
- FPG, mg/dL, mean (SD)	107.7 (6.0)	107.2 (6.3)	106.8 (5.6)	109.2 (5.9)	0.350
- HbA1c, %, mean (SD)	5.9 (0.4)	5.8 (0.4)	5.8 (0.3)	5.9 (0.4)	0.433
- Triglyceride, mg/dL, median (IQR)	128.5 (50.0, 430.0)	119.5 (88, 139)	134.5 (50.0, 430.0)	131.5 (66.0, 311.0)	0.057
- Total cholesterol, mg/dL, mean (SD)	199.1 (41.6)	189.3 (34.2)	201.9 (48.2)	207.0 (41.0)	0.319
- HDL-C, mg/dL, mean (SD)	50.8 (10.0)	51.9 (10.6)	49.7 (9.0)	50.6 (10.6)	0.738
- LDL-C, mg/dL, mean (SD)	131.8 (40.8)	126.7 (34.0)	131.6 (54.1)	138.2 (31.2)	0.628
- Hs-CRP, mg/dL, median (IQR)	1.8 (0.1, 10.3)	1.7 (0.96, 3.21)	1.3 (0.1, 6.1)	3.2 (0.8, 9.1)	0.051
- Fasting insulin, mIU/L, median (IQR)	8.7 (3.1, 38.2)	8.8 (3.1, 38.2)	8.6 (4.0, 21.7)	11.1 (3.4, 19.7)	0.595
- HOMA-IR, median (IQR)	2.29 (1.85, 3.43)	2.24 (1.83, 3.43)	2.27 (1.50, 2.93)	2.95 (1.96, 3.68)	0.540

BMI, body mass index; CAD, coronary artery disease; CKD, chronic kidney disease; CVA, cerebrovascular disease; DBP, diastolic blood pressure; DM, diabetes mellitus; FPG, fasting plasma glucose; GDM, gestational diabetes mellitus; HC, hip circumference; Hs-CRP, high-sensitivity C-reactive protein; HDL-C, high density lipoprotein cholesterol; HOMA-IR, Homeostatic model assessment of insulin resistance; IQR, interquartile range; LDL-C, low density lipoprotein cholesterol; NAFLD, non-alcoholic fatty liver disease; NC, neck circumference; SBP, systolic blood pressure; TRE, time-restricted eating; WC, waist circumference.

**Table 2 nutrients-15-04233-t002:** The mean difference in outcomes among time-restricted eating plus behavioral economic interventions, time-restricted eating alone, and usual care groups: An intention-to-treat analysis.

Outcomes.	Treatment Comparison	Mean Difference (95% CI)	*p*-Value
FPG, mg/dL	TRE vs. Usual care	−3.03 (−7.00, 0.93)	0.134
	TRE + BE vs. Usual care	−1.74 (−5.60, 2.12)	0.376
	TRE + BE vs. TRE	1.29 (−2.46, 5.04)	0.500
HbA1c, mg%	TRE vs. Usual care	−0.15 (−0.36, 0.07)	0.177
	TRE + BE vs. Usual care	−0.17 (−0.38, 0.04)	0.113
	TRE + BE vs. TRE	−0.02 (−0.21, 0.17)	0.821
DBP, mmHg	TRE vs. Usual care	0.37 (−4.04, 4.78)	0.871
	TRE + BE vs. Usual care	−3.42 (−7.70, 0.87)	0.118
	TRE + BE vs. TRE	−3.78 (−8.03, 0.47)	0.081
SBP, mmHg	TRE vs. Usual care	0.69 (−7.26, 8.63)	0.865
	TRE + BE vs. Usual care	−9.67 (−17.40, −1.95)	0.014
	TRE + BE vs. TRE	−10.36 (−17.96, −2.76)	0.008
Body weight, kg	TRE vs. Usual care	−2.67 (−8.65, 3.30)	0.381
	TRE + BE vs. Usual care	0.13 (−5.80, 6.07)	0.965
	TRE + BE vs. TRE	2.81 (−0.23, 5.85)	0.070
hs-CRP, mg/dL	TRE vs. Usual care	−1.68 (−3.20, −0.14)	0.032
	TRE + BE vs. Usual care	−0.43 (−1.91, 1.05)	0.569
	TRE + BE vs. TRE	1.25 (−0.25, 2.74)	0.103
Fasting insulin, mIU/L	TRE vs. Usual care	−1.57 (−3.37, 0.23)	0.087
	TRE + BE vs. Usual care	−1.59 (−3.34, 0.16)	0.076
	TRE + BE vs. TRE	−0.02 (−1.74, 1.70)	0.985
HOMA-IR	TRE vs. Usual care	−0.47 (−1.00, 0.05)	0.079
	TRE + BE vs. Usual care	−0.40 (−0.91, 0.11)	0.127
	TRE + BE vs. TRE	0.07 (−0.43, 0.58)	0.773
TG, mg/dL	TRE vs. Usual care	−6.32 (−25.57, 12.93)	0.520
	TRE + BE vs. Usual care	−9.35 (−28.03, 9.34)	0.327
	TRE + BE vs. TRE	−3.03 (−21.71, 15.66)	0.751
Chol, mg/dL	TRE vs. Usual care	−17.10 (−35.97, 1.78)	0.076
	TRE + BE vs. Usual care	−7.70 (−26.08, 10.69)	0.412
	TRE + BE vs. TRE	9.40 (−8.23, 27.02)	0.296
LDL-C, mg/dL	TRE vs. Usual care	−12.58 (−29.87, 4.70)	0.154
	TRE + BE vs. Usual care	−3.68 (−20.53, 13.17)	0.668
	TRE + BE vs. TRE	8.90 (−7.00, 24.79)	0.272
HDL-C, mg/dL	TRE vs. Usual care	−1.55 (−7.01, 3.91)	0.578
	TRE + BE vs. Usual care	−3.82 (−9.16, 1.52)	0.161
	TRE + BE vs. TRE	−2.27 (−7.00, 2.45)	0.347
Total energy intake, kcal/day	TRE vs. Usual care	−72.53 (−241.50, 96.44)	0.400
	TRE + BE vs. Usual care	−75.47 (−239.0, 88.05)	0.366
	TRE + BE vs. TRE	−2.94 (−166.41, 160.53)	0.972
Carbohydrate, g/day	TRE vs. Usual care	−8.36 (−33.45, 16.73)	0.514
	TRE + BE vs. Usual care	−9.09 (−33.36, 15.18)	0.463
	TRE + BE vs. TRE	−0.73 (−25.03, 23.57)	0.953
Protein, g/day	TRE vs. Usual care	−1.74 (−10.51, 7.02)	0.696
	TRE + BE vs. Usual care	−0.67 (−9.15, 7.81)	0.877
	TRE + BE vs. TRE	1.07 (−7.41, 9.55)	0.804
Total fat, g/day	TRE vs. Usual care	−3.51 (−11.77, 4.74)	0.404
	TRE + BE vs. Usual care	−4.09 (−12.08, 3.90)	0.316
	TRE + BE vs. TRE	−0.58 (−8.56, 7.41)	0.888
Cholesterol, mg/day	TRE vs. Usual care	−34.26 (−95.79, 27.26)	0.275
	TRE + BE vs. Usual care	−12.73 (−72.27, 46.80)	0.675
	TRE + BE vs. TRE	21.53 (−38.01, 81.08)	0.478
Saturated fat, g/day	TRE vs. Usual care	−2.19 (−5.00, 0.62)	0.126
	TRE + BE vs. Usual care	−2.80 (−5.51, −0.09)	0.043
	TRE + BE vs. TRE	−0.61 (−3.33, 2.12)	0.663
Simple sugar, g/day	TRE vs. Usual care	2.24 (−11.43, 15.90)	0.748
	TRE + BE vs. Usual care	−4.23 (−17.45, 8.98)	0.530
	TRE + BE vs. TRE	−6.47 (−19.72, 6.78)	0.339

CI, confidence interval; DBP, diastolic blood pressure; FPG, fasting plasma glucose; hs-CRP, high-sensitivity C-reactive protein; HDL-C, high density lipoprotein cholesterol; HOMA-IR, Homeostatic model assessment of insulin resistance; LDL-C, low density lipoprotein cholesterol; SBP, systolic blood pressure; TRE, time-restricted eating.

**Table 3 nutrients-15-04233-t003:** The mean difference in outcomes between time-restricted eating and usual care groups (as-treated analysis).

Outcomes	Mean Difference (95% CI)	*p*-Value
FPG, mg/dL	−4.74 (−8.58, −0.90)	0.015
HbA1C, mg%	−0.24 (−0.457, −0.03)	0.026
Body weight, kg	−1.01 (−7.61, 5.59)	0.765
SBP, mmHg	−7.21 (−15.25, 0.83)	0.079
DBP, mmHg	−5.28 (−9.57, −0.99)	0.016
Triglyceride, mg/dL	−18.54 (−37.03, −0.05)	0.049
Total cholesterol, mg/dL	6.30 (−13.01, 25.61)	0.523
LDL-C, mg/dL	7.77 (−9.87, 25.41)	0.388
HDL-C, mg/dL	0.16 (−5.40, 5.72)	0.954
Fasting insulin, mIU/L	−1.94 (−3.71, −0.18)	0.031
HOMA-IR	−0.61 (−1.12, −0.10)	0.020
hs-CRP	−0.92 (−2.45, 0.60)	0.236
Total energy intake, kcal/day	−182.19 (−341.46, −22.92)	0.025
Carbohydrate, g/day	−19.86 (−43.81, 4.09)	0.104
Protein, g/day	−6.81 (−15.16, 1.53)	0.110
Total fat, g/day	−8.38 (−16.22, -0.55)	0.036
Cholesterol, mg/day	−38.44 (−98.05, 21.17)	0.206
Saturated fat, g/day	−3.66 (−6.31, −1.00)	0.007
Simple sugar, g/day	−15.01 (−27.88, −2.13)	0.002

DBP, diastolic blood pressure; FPG, fasting plasma glucose; hs-CRP, high-sensitivity C-reactive protein; HDL-C, high density lipoprotein cholesterol; HOMA-IR, Homeostatic model assessment of insulin resistance; LDL-C, low density lipoprotein cholesterol; SBP, systolic blood pressure.

## Data Availability

Data described in the manuscript, code book, and analytic code will be made available upon request pending.

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
