# Peer review of "Efficacy of Time-Restricted Eating and Behavioral Economic Intervention in Reducing Fasting Plasma Glucose, HbA1c, and Cardiometabolic Risk Factors in Patients with Impaired Fasting Glucose: A Randomized Controlled Trial"

_nutrients, 2023, doi:10.3390/nu15194233_

Round 1

Reviewer 1 Report

The Authors presented very detailed analysis. The study Is well designed, results and disscusion are clearly described. However, the Authors should add more detailed information about laboratory tests in the Methods sections: what kind of methods and analyzers were used for laboratory testing and where they were performed?

Author Response

The Authors presented very detailed analysis. The study Is well designed, results and disscusion are clearly described. However, the Authors should add more detailed information about laboratory tests in the Methods sections: what kind of methods and analyzers were used for laboratory testing and where they were performed?

Response. The measurement methods have now been added in a revised section 2.5 (page 3, lines 132-141) as suggested.

Reviewer 2 Report

Major comments:

1. Please provide rationale for the diagnosis criteria of IFG as FPG of 100-125 mg/dl and HbA1c less than 6.5%. Wether those with imparied glucose tolerance is included or excluded?

2. According to the description in figure 1, 1 subject out of 26 lost follow-up in TRE+BE and 2 subjects out of 24 lost follow-up in TRE alone, however, 26 and 23 respectively finally included in ITT analysis. It seems should be 25 for TRE+BE and 22 for TRE alone.

Minor comments:

Page 4, Line 152: "and weeks 4 8, and 12 post..." should be " and weeks 4, 8, and 12 post..."

Author Response

Major comments:

  1. Please provide rationale for the diagnosis criteria of IFG as FPG of 100-125 mg/dl and HbA1c less than 6.5%. Wether those with imparied glucose tolerance is included or excluded?

Response. The definition for impaired fasting glucose in our study was based on the American Diabetes Association (ADA) Standard of Care in Diabetes 2023. This reference has been cited in the manuscript (see page 3, line 100). Our study did not measure oral glucose tolerant test. Thus, the status of impaired glucose tolerance in our participants could not be determined. We have added this point in the discussion part (see page 15).  

  1. According to the description in figure 1, 1 subject out of 26 lost follow-up in TRE+BE and 2 subjects out of 24 lost follow-up in TRE alone, however, 26 and 23 respectively finally included in ITT analysis. It seems should be 25 for TRE+BE and 22 for TRE alone.

Response. The subject in the TRE+BE group was lost to follow-up at week 12 after randomization. Therefore, this subject had outcome measures for weeks 4 and 8. One subject in the TRE alone group missed only their week 8 outcome measures; therefore, this subject had outcome measures at weeks 4 and 12. According to our accepted protocol, the missing outcome data was replaced by the last observation carried forward. Therefore, the missing outcome at week 12 for the subject from the TRE+BE group was replaced by the outcome at week 8 and the outcomes for week 8 for the subject in the TRE alone group was replaced by the outcomes at week 4. We have revised the methods for addressing missing outcome data in the updated statistical analysis section (please see page 4, line 192).  The subject lost to follow-up at week 4 in the TRE alone group was excluded from the analysis as only baseline data was available.

Minor comments:

  1. Page 4, Line 152: "and weeks 4 8, and 12 post..." should be " and weeks 4, 8, and 12 post..."

Response: Thank-you. Amended as suggested.  (please see page 4, line 160).

Reviewer 3 Report

This research article addresses the effects of time-restricted eating and behavioural economic interventions on fasting plasma glucose, HbA1c, body weight, systolic and diastolic blood pressure, fasting insulin, serum triglyceride, total cholesterol, low-density lipoprotein cholesterol, high-density lipoprotein cholesterol and high-sensitivity C reactive protein, compared with time-restricted eating alone or usual care in subjects with impaired fasting glucose in an open-label randomised controlled trial.

The authors clearly state the objective of study. However, a major limitation of this study is unpowered sample size. More revisions are required for readers to have concise information on this intervention.

In 2.5 section, authors need to address specific measurement methods (anthropometric measurement, biochemical analysis, machine info used for assessment/analysis, and so on).

In Table 3, authors only addressed intakes o total energy intake, carbohydrate, protein, total fat and cholesterol. More nutrient intakes of subjects need to be addressed.

Authors conducted an intention to treat analysis to showed the mean difference of outcomes among time restricted eating plus behavioural economic interventions, time restricted eating alone, and usual care groups, as presented Table 2 and Table 3. Authors need to explain whether outcomes of this study were obtained either at the end of the study or during whole study period for treat analysis.

Author Response

This research article addresses the effects of time-restricted eating and behavioural economic interventions on fasting plasma glucose, HbA1c, body weight, systolic and diastolic blood pressure, fasting insulin, serum triglyceride, total cholesterol, low-density lipoprotein cholesterol, high-density lipoprotein cholesterol and high-sensitivity C reactive protein, compared with time-restricted eating alone or usual care in subjects with impaired fasting glucose in an open-label randomised controlled trial.

  1. The authors clearly state the objective of study. However, a major limitation of this study is unpowered sample size. More revisions are required for readers to have concise information on this intervention.

Response. The limitation of underpowered sample size was highlighted in the discussion (page 14, line 433) and the sample size details were provided in section 2.7 to provide the reader with transparency and an opportunity to draw their own conclusions. The details of the interventions are described in the section 2.4 (page 3, lines 116-118). 

  1. In 2.5 section, authors need to address specific measurement methods (anthropometric measurement, biochemical analysis, machine info used for assessment/analysis, and so on).

Response. The measurement methods have now been added in a revised section 2.5 (page 3, lines 132-141) as suggested.

  1. In Table 3, authors only addressed intakes of total energy intake, carbohydrate, protein, total fat and cholesterol. More nutrient intakes of subjects need to be addressed.

Response. The amount of saturated fat and simple sugar intake has now been added to a revised Table 1. Comparisons of saturated fat and simple sugar intake between the three groups has also been added to revised Tables 2 and 3 and addressed in sections 3.2.1 and 3.2.2.

  1. Authors conducted an intention to treat analysis to showed the mean difference of outcomes among time restricted eating plus behavioural economic interventions, time restricted eating alone, and usual care groups, as presented Table 2 and Table 3. Authors need to explain whether outcomes of this study were obtained either at the end of the study or during whole study period for treat analysis.

Response. Both primary and secondary outcomes were obtained during (weeks 4 and 8 post randomization) and at the end of the study (week 12 post randomization). (please see page 3, line 144). The outcomes measured during and at the end of the study were considered in the analysis using a mixed-effect linear regression model by regressing outcomes (i.e., repeated measures at weeks 4, 8, and 12 post randomization) on intervention and time, considering patients as a random effect and the intervention arms as a fixed-effect (please see page 4, lines 176-182).
